# Unique Biological Characteristics of Patients with High Gleason Score and Localized/Locally Advanced Prostate Cancer Using an In Silico Translational Approach

**DOI:** 10.3390/curroncol32070409

**Published:** 2025-07-18

**Authors:** Shiori Miyachi, Masanori Oshi, Takeshi Sasaki, Itaru Endo, Kazuhide Makiyama, Takahiro Inoue

**Affiliations:** 1Department of Nephro-Urologic Surgery and Andrology, Mie University Graduate School of Medicine, Tsu 514-8507, Mie, Japan; s-miyachi@med.mie-u.ac.jp (S.M.); tinoue28@med.mie-u.ac.jp (T.I.); 2Department of Gastroenterological Surgery, Yokohama City University Graduate School of Medicine, Yokohama 236-0004, Kanagawa, Japan; oshi.mas.wc@yokohama-cu.ac.jp (M.O.); endoit@yokohama-cu.ac.jp (I.E.); 3Department of Urology, Yokohama City University Graduate School of Medicine, Yokohama 236-0004, Kanagawa, Japan; makiya@yokohama-cu.ac.jp

**Keywords:** prostate cancer, Gleason score, gene expression, GSVA, xCell

## Abstract

Gleason score (GS) is one of the best predictors of prostate cancer aggressiveness. GS classifies cancer cells based on the histological patterns of prostate tissue sections and does not evaluate the nuclear grade or proliferation of the cancer cells. A small number of studies focused on the association between gene expression signatures and GS using an in silico translational approach. GS is associated with cell cycle-related genes, changes in DNA repair genes, stromal-related genes, immune-related genes, cuprotosis-related genes, and several specific genes. To our knowledge, this is the first report showing that GS was positively correlated not only with homologous recombination deficiency mutations but also with intratumor heterogeneity, fractional mutations, single nucleotide variant neoantigen, silent mutation rate, and non-silent mutation rate. Our findings emphasize that GS reflects not only morphological abnormalities but also differences in cancer cell proliferation, immune cell infiltration, and high mutation rates, which may affect prognosis.

## 1. Introduction

Prostate cancer (PCa) is one of the most commonly diagnosed cancers worldwide. It is the fourth most diagnosed and eighth leading cause of death, with 1,467,854 diagnoses and 397,430 deaths in 2022, according to the World Health Organization (available at https://gco.iarc.who.int/today; accessed on [4 April 2024]). Partly because of prostate-specific antigen (PSA) screening, most patients with PCa have localized cancers [1,2]. Patients with nonmetastatic localized/locally advanced PCa generally receive curative treatment with radical prostatectomy (RP) or radiation therapy (RT). After RP or RT, certain patient subgroups with poor tumor differentiation, i.e., high Gleason scores (GSs), have high rates of biochemical recurrence (BCR), the development of distant metastases, and cancer-specific mortality [3].

Among the pathological features based on biopsy or RP specimens, GS is one of the best predictors of PCa aggressiveness in patients with localized/locally advanced PCa undergoing RP or RT [3]. The Gleason grading system, developed by Dr. Donald Gleason in 1966, remains the basis for PCa management [4]. Notably, it classifies cancer cells based on the histological patterns of their arrangement in hematoxylin and eosin-stained prostate tissue sections [5] and does not evaluate the nuclear grade or proliferation of the cancer cells. In accordance with the 2014 International Urologic Pathology Association Consensus Conference, the current grade groups based on the Gleason grading system define GG1 (Grade Group 1), i.e., Gleason Score ≤ 6; GG2, i.e., Gleason Score 3 + 4 = 7; GG3, i.e., Gleason Score 4 + 3 = 7; GG4, i.e., Gleason Score 8; and GG5, i.e., Gleason Score 9–10 [6]. Various prognostic factors of RP specimens (i.e., extraprostatic extension, lymphovascular space invasion, and seminal vesicle invasion) have been examined and reported in addition to the Gleason grading system [7] but there are still no methods relevant to the prognosis of RP specimens beyond the Gleason grading system. Furthermore, the Gleason grading system is an important prognostic factor not only for localized PCa but also for metastatic PCa treated with second-generation androgen receptor axis-targeted agents, the most current agents of this era [8]. Thus, the elucidation of the molecular mechanism of the Gleason grading system, with its universal significance, can help fight the root causes of PCa.

Prior to the clinical use of poly-ADP ribose polymerase (PARP) inhibitors (Olaparib, Talazoparib, Niraparib, and Rucaparib) [9], the treatment agents for metastatic PCa were androgen deprivation therapy (Abiraterone, Enzalutamide, Apalutamide, and Darolutamide) or cytotoxic anti-cancer agents (Docetaxel, Cabazitaxel). PARP inhibitors are highly anticipated as a treatment for progressive PCa based on pathogenic mutations in genes, especially *BRCA1* and *BRCA2*, and heralded the dawn of personalized medicine in this field. DNA is damaged by cellular metabolites, exposure to environmental agents, and chemical bonds that spontaneously decay under physiological conditions, and it is constantly being repaired. In both normal and cancer cells, multiple DNA damage response pathways are relied upon to repair various forms of DNA damage. PARP binds to single-strand DNA breaks and plays an important role in DNA repair. If single-strand DNA break repair fails for some reason, the process shifts to double-strand breaks. In tumor cells with *BRCA* mutations, damaged DNA cannot be repaired by homologous recombination, leading to cell death. PARP inhibitors exploit this mechanism to induce tumor cell death by inhibiting single-strand DNA break repair in tumor cells with *BRCA* mutations [10]. Thus, PARP inhibitors are drugs that take advantage of the phenomenon that a single gene defect is not lethal to cells but the coexistence of two gene defects is lethal (synthetic lethality). Homologous recombination deficiency is a condition caused by loss-of-function mutations or the epigenetic inactivation of homologous recombination repair (HRR) genes (e.g., *BRCA1* and *BRCA2*), which is called BRCAness [11], which prevents accurate repair, and it is inferred that homologous recombination deficiency (HRD) could be more sensitive to PARP inhibitors. Identifying the association between HRD and GS could be useful for predicting the efficacy of PARP inhibitors.

The development of basic research technology has been remarkable, and the number of research articles on PCa using sequencing and omics technology has increased dramatically [12]. Representative sequencing methods include next-generation sequencing, RNA sequencing, and protein sequencing. Some genomic studies have focused on the Y chromosome, investigating its association with PCa and attempting to clarify the correlation between the loss of the Y chromosome and PCa development [13]. Recently, the number of single-cell RNA sequencing research articles focusing on PCa microenvironments has especially increased [14,15,16]. These sequencing and omics analyses have been performed on a variety of targets (i.e., genomics, epigenomics, transcriptomics, proteomics, metabolomics, and multi-omics) and could result in the identification of new therapeutic targets, diagnostic markers, and prognostic factors by elucidating the molecular pathogenesis of PCa [12]. We believe that understanding the molecular mechanisms of the Gleason grading system as it relates to diagnosis, treatment, and prognosis is the most critical unsolved issue in PCa.

A small number of studies focused on the association between gene expression signatures and GS using an in silico translational approach [17,18,19,20,21,22,23,24]. These studies demonstrated that GS is associated with cell cycle-related genes [17,22], DNA repair gene alterations [18,22], stromal-related genes [19], immune-related genes [22], cuprotosis-related genes [23], and some specific genes [20,21,22,24]. To characterize the biology of GS, it is important to assess the state of the tumor microenvironment (TME) and cancer cells [25]. In addition, several studies have documented the expression of diverse genes and signaling pathways in PCa. However, it is difficult to clearly distinguish GSs with single or multiple gene expressions in studies. Interactions of these gene expressions are intricate and foster cancer cell proliferation and resistance to treatment. Hence, defining complex signaling solely through a single gene can pose challenges. Gene Set Variation Analysis (GSVA) is a valuable tool for comprehensively elucidating cancer signaling pathways [26]. Previously, we reported that various gene set pathway scores correlated with clinical outcomes in other cancers using the GSVA methodology, and scoring by GSVA can be a more powerful biomarker than individual genes [27]. In addition, we reported that the xCell algorithm, a gene signature-based method for inferring 64 immune and stromal cell types, estimated the fractions of several immune and stromal cell types in the TME [28].

Here, using GSVA and xCell analyses of a large PCa patient cohort in the datasets of The Cancer Genome Atlas (TCGA) and Gene Expression Omnibus (GEO), we hypothesized that patients with PCa with high GS levels would have unique cancer biology characteristics, including a TME associated with poor outcomes.

## 2. Materials and Methods

### 2.1. Data Acquisition for Patients with Localized/Locally Advanced PCa

To investigate the molecular characteristics and clinical implications of GS in localized and locally advanced PCa, we curated a total of 741 tumor samples with corresponding transcriptomic and clinicopathological data from two widely recognized public databases. We utilized The Cancer Genome Atlas (TCGA) (PRAD; *n* = 493, RP cohort), accessible via cBioPortal (https://www.cbioportal.org/study/summary?id=prad_tcga_pan_can_atlas_2018 (accessed on 15 Februay 2024 )). In parallel, we incorporated a second independent cohort, derived from the Gene Expression Omnibus (GEO) under accession number GSE116918 (*n* = 248, RT cohort) [29]. For both datasets, we utilized normalized gene expression data, which had been log2-transformed to facilitate downstream statistical modeling and reduce heteroscedasticity. The RP cohort provided longitudinal data on recurrence-free survival and overall survival (OS), enabling the analysis of long-term oncologic outcomes. The RT cohort, by contrast, included information on biochemical recurrence-free survival (BCR-FS) and metastasis-free survival (MFS), thus complementing our investigation by enabling the evaluation of different therapeutic contexts. These two datasets were deliberately chosen to ensure the robustness and generalizability of our findings across clinical scenarios.

### 2.2. Pathway Enrichment Analysis via GSVA

To assess the activity of specific molecular pathways across tumor samples, we implemented the GSVA algorithm, an unsupervised, non-parametric method that quantifies the enrichment of predefined gene sets in individual samples [26]. GSVA enables the estimation of pathway-level activation in bulk tumor transcriptomes by summarizing the coordinated expression of gene sets derived from curated databases (MSigDB). The resulting GSVA scores, computed for each pathway in each patient sample, serve as proxies for underlying biological activity rather than individual gene expression [26].

### 2.3. Estimation of Tumor Microenviornment Components

Given the growing recognition of the TME in modulating cancer progression, we aimed to characterize the immune and stromal cell landscape associated with GS patterns. Fo this purpose, we applied xCell algorithm [30], a computational framework that infers the relative abundance of 64 immune and stromal cell types from transcriptomic profiles. This approach integrates gene signatures reflective of specific cell populations and corrects for potential spillover effects between similar cell types.

### 2.4. Assessment of Genomic Instability and Mutation Burden

To further dissect the biological underpinnings associated with GS, we evaluated indicators of genomic instability and mutational processes, with a particular focus on HRD-related parameters. These analyses were limited to the TCGA RP cohort, which offers comprehensive whole-exome sequencing data. Genomic features including the fraction of the genome altered, single nucleotide variant (SNV) burden, insertion–deletion (indel) neoantigen load, and counts of silent and non-silent mutations were calculated using the methodology described by Thorsson et al. [31]. This allowed us to probe potential associations between pathway activity, mutational processes, and DNA repair deficiencies.

### 2.5. Statistical Analysis

All statistical analyses were performed using R software (version 4.1.0), ensuring reproducibility and transparency. Quantitative data distributions were visualized using Tukey-style boxplots, which display medians and interquartile ranges (IQRs), with whiskers extending to 1.5 times the IQR. For comparisons across more than two groups, we employed the Kruskal–Wallis test, a non-parametric alternative to ANOVA suitable for skewed data distributions. Survival analyses were conducted using the Kaplan–Meier method, and differences in survival curves were assessed by the log-rank test. Statistical significance was defined as a two-tailed *p*-value < 0.05 unless otherwise stated.

## 3. Results

### 3.1. Patients Characteristics

The clinical characteristics of the patients in the TCGA and GEO (GSE116918) cohorts are shown in Table 1. The group of patients who received radical prostatectomy had a younger median age than the group who received radiation therapy, while the group who received radiation therapy tended to have more locally advanced clinical T3-stage cancer.

### 3.2. GS Levels Were Associated with Patient Prognosis in Localized/Locally Advanced PCa

We analyzed the associations between patient prognosis and GS levels in localized/locally advanced PCa. High GS levels were associated with poor outcomes for BCR-FS and OS in the TCGA cohort (both *p* < 0.001) (Figure 1a) and for BCR-FS and MFS in the GSE116918 cohort (*p* = 0.281 and 0.247, respectively) (Figure 1b). The TCGA cohort showed significant differences in high GS levels and prognosis, while the GSE116918 cohort showed a trend toward poor prognosis in high GS levels, although not significantly. Possible reasons for the lack of significant differences in BCR-FS and MFS in the radiation therapy cohort could be that the follow-up period was not long enough or the co-administration of androgen deprivation therapy.

### 3.3. GS Levels Were Correlated with the Activity Levels of Cell Proliferation-Related Gene Sets

We hypothesized that high GS levels in PCa would lead to more aggressive molecular cancer biology. In order to test our hypothesis, we conducted a gene set enrichment analysis and found that high GS levels in PCa enriched all of the cell proliferation-related Hallmark gene sets, including E2F targets, the G2M checkpoint, the mitotic spindle, and MYC targets v1 and v2. GS levels were positively correlated with the activity levels of cell proliferation-related gene sets, including E2F targets, the G2M checkpoint, the mitotic spindle, and MYC targets v1 and v2, consistently in the TCGA and GSE116918 cohorts (all *p* < 0.001) (Figure 2).

### 3.4. GS Levels Were Associated with the Activity Levels of Immunity-Related Gene Sets

Since immune cells in the TME play a critical role in cancer progression and response to treatments, we investigated the association of GS levels with several immune response-related pathways including inflammatory response, interferon (IFN)-α and IFN-γ responses, allograft rejection, IL2/STAT5 signaling, IL6/JAK/STAT3 signaling, coagulation cascade, and complement signaling. Although there was a positive correlation between GS levels and activity levels of several immune-related gene sets, such as interferon-α response (*p* = 0.026), interferon-γ response (*p* = 0.002), allograft rejection (*p* < 0.001), IL2/STAT5/signaling (*p* < 0.001), and complement (*p* = 0.044) in the GSE116918 cohort, no significant difference was observed in the TCGA cohort (Figure 3).

### 3.5. GS Levels Were Associated with the Infiltration Fraction of Several Immune Cells in the TME of PCa

We investigated the association between GS levels and immune cell infiltration in the TME. GS levels were positively correlated with the infiltration fraction of several anti-cancer immune cells, including CD4+ memory T cells, dendritic cells (DCs), macrophage M1, and pro-cancerous T helper type 2 (Th2) cells, consistently in both the TCGA and GSE116918 cohorts (all *p* < 0.05) (Figure 4).

### 3.6. GS Levels Were Not Associated with the Infiltration Fraction Rate of Stromal Cells

We investigated the association between GS levels and stromal cell infiltration in the TME. GS levels were not associated with the infiltration fraction rate of stromal cells, including fibroblasts, adipocytes, endothelial cells, and pericytes, in both the TCGA and GSE116918 cohorts (Figure 5).

### 3.7. GS Levels Were Associated with the Score Levels of Homologous Recombination Defeciency (HRD), Intratumor Heterogeneity, Fraction Alteration, and Mutation Load

Since cancer cell growth is associated with high mutation rates, it is interesting to examine the relationship between GS levels and mutation rates in PCa. HRD are representative of defects in DNA repair and are surrogate markers of increased mutational load. We found that GS levels were positively associated with the score levels of HRD, intratumor heterogeneity, fraction alteration, SNV neoantigens, silent mutation rate, and non-silent mutation rate in the TCGA cohort (all *p* < 0.001) (Figure 6).

## 4. Discussion

In this study, we explored the biological characteristics of GS in PCa using in mRNA analysis. This study demonstrated that PCa with high GS levels was enriched for all five hallmark cell proliferation-related gene sets (E2F targets, the G2M checkpoint, the mitotic spindle, and MYC targets v1 and v2) and was significantly associated with high levels of the infiltrating fraction of several immune cells (CD4+ memory T cells, DCs, macrophage M1, and Th2 cells). Notably, to our knowledge, this is the first report showing that based on mRNA expression analysis, GS was positively correlated not only with HRD mutations but also with intratumor heterogeneity, fractional mutations, SNV neoantigens, silent mutation rate, and non-silent mutation rate. These observations emphasize that GS reflects not only morphological abnormalities but also differences in cancer cell proliferation, immune cell infiltration, and high mutation rates, which may affect prognosis. Furthermore, recent advances in cancer biology highlight the importance of transcriptional signatures as integrated readouts of tumor signaling states, beyond individual gene mutations. For example, in the concept of “BRCAness,” gene expression patterns can reveal homologous recombination deficiencies even in the absence of *BRCA* mutations [32]. Similarly, mRNA profiles often reflect the complex interplay of signaling pathways more effectively than isolated protein markers. In this study, we focused on transcriptomic analysis to capture such dynamic alterations. We observed a clear and consistent increase in the expression of cell proliferation-related gene sets with rising Gleason scores consistently in two large independent cohorts, suggesting that mRNA expression levels can serve as a sensitive indicator of tumor aggressiveness.

Several mechanisms have been reported to be involved in GS and cell proliferation based on gene expression analysis [17,21,22,33]. For instance, cell cycle-related genes (BUB1, CENPE, CENPF, DLGAP5, PRC1, and SMC4), cell cycle-related pathways (G2M and E2F), and mitosis-related genes (BUB1B and NCAPG) were correlated with GS in the TCGA database [17,21]. Moreover, the Gleason grade groups are correlated with increased cell division, telomere lengthening, and DNA damage [22]. In contrast, patients with low GS show antitumor molecular changes related to metastasis and proliferation [33]. In our study, we confirmed that GS levels were positively correlated with the activity levels of cell proliferation-related gene sets using the GSVA algorithm, consistently with multiple large independent cohorts. The consistent results across cohorts with diverse backgrounds indicate the robustness of our findings.

RNA sequencing data in this study showed a correlation between genes involved in cell proliferation and GS, but the evaluation at the protein expression level (Ki67 labeling index) of prostate cancer and cell proliferation is controversial. Ki67 is a nuclear antigen and is expressed on proliferating cells from the G1 to the M phase of the cell cycle. The Ki67 labeling index is a representative method of assessing cell proliferation by immunostaining, and the Ki67 labeling index is used to assess invasive cancer in breast cancer, but not for diagnosis in prostate cancer. In a recent article evaluating the Ki67 labeling index using artificial intelligence, the histopathological evaluation of radical prostatectomy specimens showed that the Ki67 labeling index increased as GS increased, but GSs of 8 or higher were used and the standard deviation was large and inaccurate [34]. Moreover, in the article on the Ki67 labeling index specific to GSs of 8 or higher, the inter- and intra-tumor variability of the Ki67 labeling index in high-grade PCa was shown, with a surprisingly low Ki67 labeling index in a subset of cases [35].

Since PARP inhibitors have been used in clinical practice [9], DNA repair mutations, especially those that regulate HRD, have received much attention; however, there are only a few reports on the relationship between GS and DNA repair mutations [18,22]. Marshall et al. reported the prevalence of gene alterations in the DNA repair pathway, especially HRD, in localized PCa [18]. DNA repair mutations are more prevalent in men with higher Gleason grades (group 3 and higher) [18]. Notably, our study based on mRNA expression analysis using Thorsson’s method [31] revealed that GS was positively correlated not only with HRD mutations but also with intratumor heterogeneity, fraction alteration, SNV neoantigens, silent mutation rates, and non-silent mutation rates. Our findings suggested that PARP inhibitors may be effective in patients with high GS levels but also suggested that patients with high GS levels may be resistant to medication treatments due to intratumor heterogeneity and that immune checkpoint inhibitors might be effective due to neoantigens in patients with high GS levels. However, there is currently no evidence that PARP inhibitors and immune checkpoint inhibitors are effective in high GS levels.

Ongoing clinical trials are evaluating the efficacy of immunotherapy in PCa. Recently, several studies have reported a correlation between GS and immune-related gene signatures [19,20,22,36]. The CIBERSORT and ESTIMATE analysis tools have been used for immuno-related gene analyses [19,20,22,36], but in our analysis, we used the xCell algorithm, which can evaluate a larger number of TME cells. Our results showed that PCa with high GS levels was significantly enriched with the relative abundance of immune-related gene sets, such as interferon-α and -γ responses, and associated with high levels of the infiltrating fraction of CD4+ memory T cells, DCs, macrophage M1, and Th2 cells. In the present study, interferon-α and -γ responses tended to correlate with GS, and they may have anti-tumor effects in PCa with a high GS tumor status. Among anti-cancerous immune cells, CD4+ memory T cells, DCs, macrophage M1, and Th2 cells are significantly higher in high-grade tumors (high GS levels) because high-grade cancers attract host anti-tumor immune cells and it is assumed that this is because host anti-tumor immune cells gather in highly malignant cancers. On the other hand, pro-cancerous immune cells show little tendency to correlate with GS. Yimamu et al. also demonstrated that tumors with high GS levels were significantly correlated with high fractions of CD4 T cell memory resting [20]. However, correlations between GS and immune cells (M0 macrophages, M2 macrophages, B cells, T cells, DCs, and neutrophils) based on RNA-seq data have not yielded consistent results, perhaps because of differences in analysis methods [20,22]. Moreover, it is important to note when interpreting reports using RNA-seq databases that the data do not confirm immunological changes over time but only capture the immunological status of patients at that time. Therefore, it is thought that a state in which both pro-cancerous and anti-cancer immune cells are active could be represented. Immunohistological studies of immune cells are also needed as a future project.

In addition to immune cells, the TME is composed of infiltrating fibroblasts, adipocytes, endothelial cells, and pericytes. Previous studies have shown a positive correlation between stromal score and GS [19,20]; however, in our analysis using the xCell algorithm, there was no correlation between stromal cells (fibroblasts, adipocytes, endothelial cells, and pericytes) and GS. Even if there are no differences in the infiltration fraction rate of stromal cells, stromal cells cannot necessarily be evaluated by infiltrating fraction rates because fibroblasts can change their properties into cancer-associated fibroblasts [25]. Another possible reason why there were no differences in stromal cells stratified by GS levels is that the data analyzed in this study were not necessarily extracted from only the tissue near the tumor but may have included some cells distant from the tumor. In single-cell analysis, 12 senescence-related genes in TME were correlated with GS, suggesting that the surrounding stromal cells may have reacted in response to senescence [15].

This study had several limitations. First, this was a retrospective study using previously published cohorts with some missing clinical data and treatment details and was thus susceptible to selection bias. We were able to show similar results with the two datasets in this study; however, validation with other external or the author’s own datasets was lacking. While bioinformatics analysis using a public database of human tumor samples, which was the approach used in this study, has the great advantage of being validated on multiple human samples, it has several drawbacks. For example, the quality and consistency of the data may vary from dataset to dataset, and sample-to-sample variability and noise may affect the results. In addition, these data are often collected from different institutions, with different protocols and conditions, so the results of the analyses are not always consistent. However, even with these limitations, we believe that our study was a remarkable achievement, as we were careful in interpreting the results and focused on the fact that we were able to confirm the reproducibility of our results using multiple databases. Second, the analyses in this study only showed a single-time-point “snapshot” and were not suitable for determining detailed mechanisms. It is important to examine these mechanisms in vivo/vitro. However, this study included the complex microenvironment within human tumors, and it is difficult to reproduce the human tumor microenvironment in preclinical models at this time. We believe that a further similar study would require creating organoids from human tumor tissue matched to each GS and comparing the growth potential of the tumors. Further comparative analyses through prospective studies are required to establish causal relationships and elucidate the underlying mechanisms.

The methodology (GSVA, xCell, and Thorsson’s method [31]) we used in this study have the advantage that the Gleason grading system can be compared as a gene cluster. This in silico study revealed that high GS levels were associated with cancer cell proliferation, immune cell infiltration, and high mutation rates, which may reflect worse clinical outcomes in patients with localized/locally advanced PCa. Using the analytical methods of this study, it will be possible to clarify the relationship between clinical parameters (PSA values, clinical stage, etc.) and cell biology in the TME.

## 5. Conclusions

To our knowledge, this is the first report showing that based on mRNA expression analysis, GS was positively correlated not only with HRD mutations but also with intratumor heterogeneity, fractional mutations, SNV neoantigens, silent mutation rate, and non-silent mutation rate. These observations emphasize that GS reflects not only morphological abnormalities but also differences in cancer cell proliferation, immune cell infiltration, and high mutation rates, which may affect prognosis. Our findings also suggest that PARP inhibitors may be effective in patients with high GS levels but also suggest that patients with high GS levels may be resistant to medication treatments due to intratumor heterogeneity and that immune checkpoint inhibitors might be effective due to neoantigens in patients with high GS levels. The efficacy of PARP inhibitors and immune checkpoint inhibitors in patients with high GS levels needs to be validated by future clinical studies.

## Figures and Tables

**Figure 1 curroncol-32-00409-f001:**
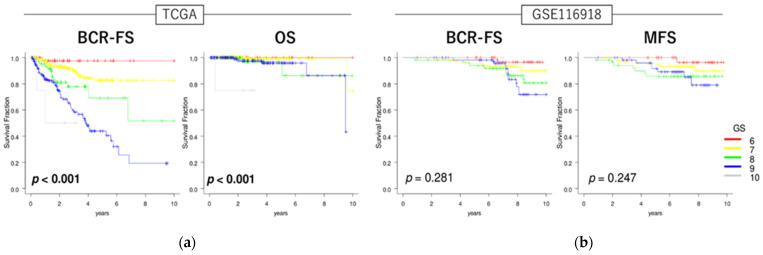
Clinical relevance of GS levels in localized/locally advanced PCa. Kaplan–Meier curve with log-rank *p*-values of BCR-FS and OS in TCGA cohort (**a**) and BCR-FS and MFS in GSE116918 cohort (**b**) of patients with localized/locally advanced PCa stratified by GS levels.

**Figure 2 curroncol-32-00409-f002:**
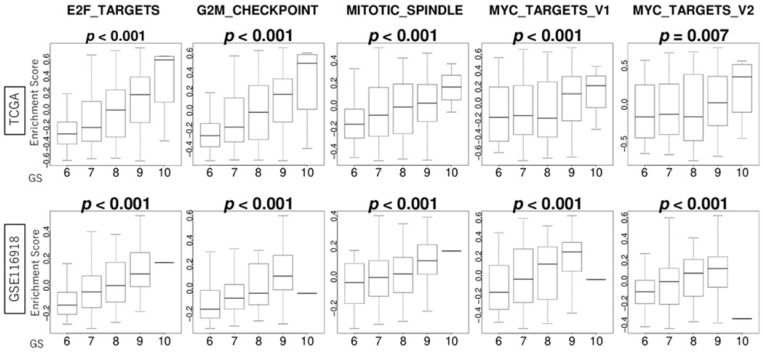
Association of GS levels with the activity levels of cell proliferation-related gene sets in PCa. Boxplots of the score levels of E2F targets, the G2M checkpoint, the mitotic spindle, and MYC targets v1 and v2 by GS levels in PCa in the TCGA and GSE116918 cohorts.

**Figure 3 curroncol-32-00409-f003:**
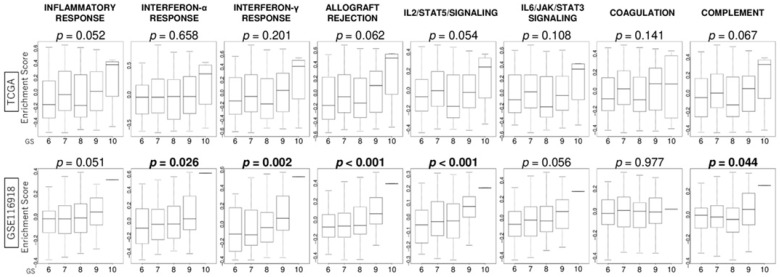
Association of GS levels with the activity levels of immunity-related gene sets in PCa. Boxplots of the score levels of immune-related gene sets including inflammatory response, interferon-α response, interferon-γ response, allograft rejection, IL2/STAT5/signaling, IL6/JAK/STAT3/signaling, coagulation, and complement in the GSE116918 cohort.

**Figure 4 curroncol-32-00409-f004:**
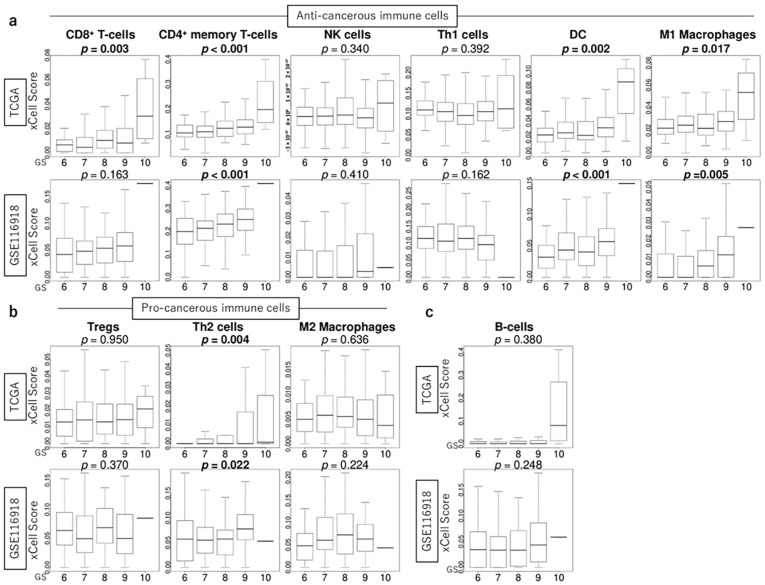
Association of GS levels with the infiltration fraction of several cells in the tumor microenvironment using the xCell algorithm. Boxplots, according to GS levels in the TCGA and GSE116918 cohorts, of the score levels of anti-cancerous immune cells, including CD8+ cells, CD4+ memory T cells, NK cells, Th1 cells, DCs, and M1 macrophages; pro-cancerous immune cells, including Tregs, Th2 cells, and macrophages M2; and B cells.

**Figure 5 curroncol-32-00409-f005:**
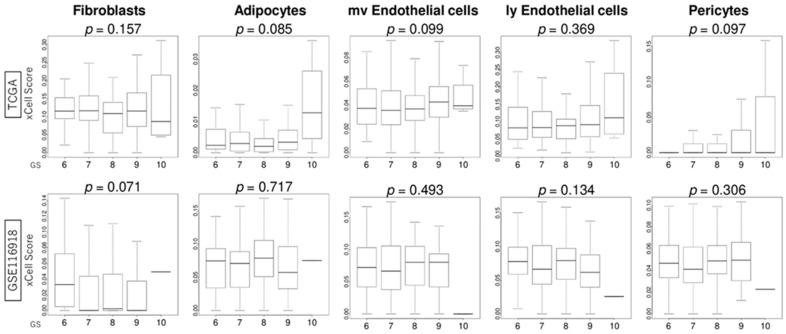
Association of GS levels with the score levels of the infiltration fraction of stromal cells in the tumor microenvironment using the xCell algorithm. Boxplots of the score levels of fibroblasts, adipocytes, microvascular (mv) and lymphatic (ly) endothelial cells, and pericytes according to GS levels in the TCGA and GSE116918 cohorts.

**Figure 6 curroncol-32-00409-f006:**
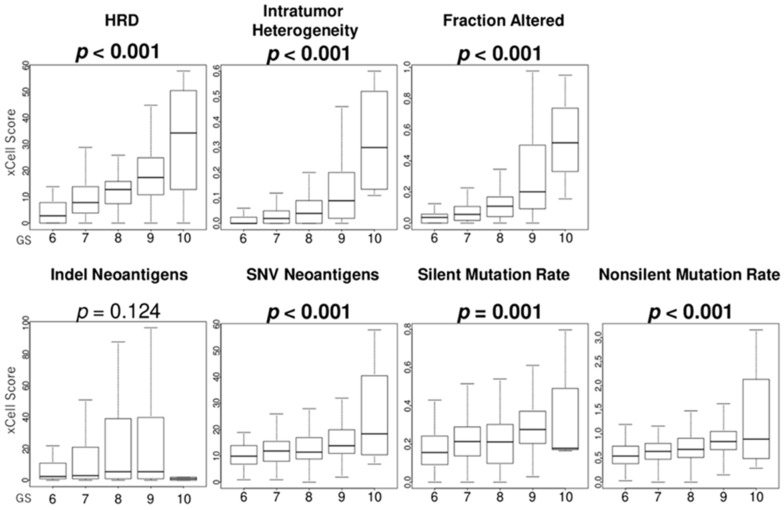
Association of GS levels with the score levels of homologous recombination deficiency (HRD), intertumoral heterogeneity, fraction alteration, and mutation load-related gene sets. Boxplots of the score levels of HRD, intratumor heterogeneity, fraction alteration, indel neoantigens, SNV neoantigens, silent mutation rates, and non-silent mutation rates in the TCGA cohort.

**Table 1 curroncol-32-00409-t001:** Descriptive characteristics of patients with localized/locally advanced prostate cancer in TCGA (*n* = 493, radical prostatectomy cohort) and GSE116918 (*n* = 248, radiation therapy cohort).

Cohort	TCGA	GSE116918
Treatment	Radical prostatectomy	Radiation therapy
Number of patients	493	248
Median age (range) years at surgery	61 (41–78)	68 (48–79)
Clinical T stage at diagnosis (%)		
cT1/T2/T3/T4/Unknown	175/172/52/2/92 (35.5/35/10/0.5/19)	51/76/92/4/25 (21/31/37/1/10)
Biopsy GS (%)		
6	45 (9)	42 (17)
7	245 (50)	99 (40)
8	62 (12)	52 (21)
9	137 (28)	54 (21.5)
10	4 (1)	1 (0.5)

## Data Availability

The datasets generated during and/or analyzed during the current study are available through TCGA (https://www.cbioportal.org/study/summary?id=prad_tcga_pan_can_atlas_2018 (accessed on 15 Februay 2024) and GEO (GSE116918).

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
