# Peer review of "Unique Biological Characteristics of Patients with High Gleason Score and Localized/Locally Advanced Prostate Cancer Using an In Silico Translational Approach"

_curroncol, 2025, doi:10.3390/curroncol32070409_

Round 1

Reviewer 1 Report

Comments and Suggestions for Authors

In this work, Miyachi S et al. show that Gleason score (GS) was positively correlated not only with homologous recombination deficiency mutations but also with intratumor heterogeneity, fractional mutations, single nucleotide variant neoantigen, silent mutation rate, and non-silent mutation rate. Thus, they demonstrate that GS reflects not only morphological abnormalities but also differences in cancer cell proliferation, immune cell infiltration, and high mutation rates, which may affect prognosis.

Major points

  1. The authors should clarify in the title of the work that it is an exclusively in silico analysis.

  1. In Table 1, the authors should discuss the differences found between patients undergoing radical prostatectomy and radiation therapy.

  1. I would strongly support the results obtained if the authors worked with some prostate tumor samples, in which, for example, they evaluated their proliferative activity (on the tissue itself, or on primary culture of cells from those tumors). It is known that mRNA expression does not always correlate with protein expression. Therefore, it would be necessary for the authors to be able to support their results by measuring at least some of the proteins in which they observe changes in mRNA expression.

  1. In the Results section, 3.5, where the authors investigate the association between GS levels and immune cell infiltration in the TME, it would strengthen the obtained results if the authors evaluated at least in some prostate cancer tissues the actual infiltration of immune cells.

  1. In the Discussion. In recent years, many working groups have demonstrated the importance of the dialogue between the tumor and the tumor microenvironment (surrounding stromal cells and extracellular matrix). The authors should discuss this point, particularly given their analysis of stromal cells and the immune infiltrate.

Minor points

The graphics in Figure 1 are out of place. They need to be realigned.

Author Response

Reviewer 1

Comments and Suggestions for Authors

In this work, Miyachi S et al. show that Gleason score (GS) was positively correlated not only with homologous recombination deficiency mutations but also with intratumor heterogeneity, fractional mutations, single nucleotide variant neoantigen, silent mutation rate, and non-silent mutation rate. Thus, they demonstrate that GS reflects not only morphological abnormalities but also differences in cancer cell proliferation, immune cell infiltration, and high mutation rates, which may affect prognosis.

Major points

  1. The authors should clarify in the title of the work that it is an exclusively in silico analysis.

 We thank the reviewer’s comment. In accordance with the reviewer's instructions, the words “using an in silico translational approach” have been added to the title.

  1. In Table 1, the authors should discuss the differences found between patients undergoing radical prostatectomy and radiation therapy.

We thank the reviewer’s comment. The group of patients who received radical prostatectomy had a younger median age than the group who received radiation therapy, while the group who received radiation therapy tended to have more locally advanced clinical T3 stage cancer. We had added a sentence in the Results section of page 4, lines 183-185.

  1. I would strongly support the results obtained if the authors worked with some prostate tumor samples, in which, for example, they evaluated their proliferative activity (on the tissue itself, or on primary culture of cells from those tumors). It is known that mRNA expression does not always correlate with protein expression. Therefore, it would be necessary for the authors to be able to support their results by measuring at least some of the proteins in which they observe changes in mRNA expression.

We appreciate the reviewer’s insightful comments. As the reviewer pointed out, mRNA expression levels do not always correlate with protein abundance or activity, and protein-level validation is indeed an important consideration. While Ki-67 is a widely used proliferation marker at the protein level, its clinical utility in prostate cancer remains limited. For instance, Blessin et al. (J Pathol. 2023 May;260(1):5-16.) demonstrated increased Ki-67 labeling index with higher Gleason scores using AI-assisted quantification, but also reported substantial variability, particularly in Gleason ≥8 cases. Similarly, Vlajnic et al. (Pathobiology. 2022;89(2):74-80.) reported marker inter- and intra-tumoral heterogeneity in Ki-67 expression among high-grade tumors (Discussion section of page 10, lines 199-311). Furthermore, recent advances in cancer biology highlight the importance of transcriptional signatures as integrated readouts of tumor signaling states, beyond individual gene mutations. For example, in the concept of “BRCAness,” gene expression patterns can reveal homologous recombination deficiencies even in the absence of BRCA mutations (Cells. 2022 Dec 1;11(23):3877). Similarly, mRNA profiles often reflect the complex interplay of signaling pathways more effectively than isolated protein markers. In this study, we focused on transcriptomic analysis to capture such dynamic alterations. We observed a clear and consistent increase in the expression of cell proliferation-related gene sets with rising Gleason scores consistently in two large independent cohorts, suggesting that mRNA expression levels can serve as a sensitive indicator of tumor aggressiveness. In light of these limitations, we have emphasized the RNA-level findings in this study and noted in the manuscript that the findings are worth reporting (Discussion section of pages 9-10, lines 277-287).

  1. In the Results section, 3.5, where the authors investigate the association between GS levels and immune cell infiltration in the TME, it would strengthen the obtained results if the authors evaluated at least in some prostate cancer tissues the actual infiltration of immune cells.

We thank the reviewer for this valuable suggestion. As the reviewer pointed out, validating immune cell infiltration in actual prostate cancer tissues would indeed strengthen our findings. However, prostate cancer is typically characterized by a "cold " tumor microenvironment, with low levels of T cell infiltration and limited immune activation. Given this baseline suppression, direct immunohistochemical validation in limited tissue samples may not adequately capture the complexity or temporal dynamics of immune cell interactions in situ. Furthermore, our analysis represents a transcriptional snapshot at a single time point, which, while informative for identifying patterns of immune-related gene expression, is inherently limited in resolving mechanistic or spatial features of immune cell localization. We fully agree that tissue-level validation of immune infiltration will be essential to extend our findings, and we have now included a statement in the discussion noting that further immunohistochemical studies are planned as part of future investigations (Discussion section of page 11, lines 348-349).

  1. In the Discussion. In recent years, many working groups have demonstrated the importance of the dialogue between the tumor and the tumor microenvironment (surrounding stromal cells and extracellular matrix). The authors should discuss this point, particularly given their analysis of stromal cells and the immune infiltrate.

We thank the reviewer for this important suggestion. In light of our analysis of stromal and immune cell infiltration, we have expanded the discussion to address the role of tumor microenvironment interactions.

Our findings support the view that high-grade prostate cancers exist within a dynamic immune microenvironment. We observed that interferon-α and -γ responses tended to correlate with GS, suggesting an activation of anti-tumor immune signaling in more aggressive tumors. Consistently, several immune cell subsets known for their anti-tumor roles, including CD4+ memory T cells, DC, macrophage M1, and Th2 cells, were significantly higher infiltrated in high GS. This may reflect the recruitment of host immune effector cells in response to increased tumor immunity. On the other hand, pro-cancerous immune cells did not show a clear association with GS (Discussion section of page 11, lines 333-339).

Regarding stromal cells, we did not observe a significant difference in stromal cell abundance when stratified by GS level. One possible explanation is that stromal cells cannot necessarily be evaluated by infiltrating fraction rates because fibroblasts can change their properties into cancer-associated fibroblasts, we assume that that the data analyzed in this study were not necessarily extracted from only the tissue near the tumor, but may have included some cells distant from the tumor (Discussion section of page 11, lines 356-359). We have incorporated these points into the discussion section to better contextualize the role of the tumor microenvironment in our findings.

Minor points

The graphics in Figure 1 are out of place. They need to be realigned.

We thank the reviewer’s comment. We have rearranged the Figure 1.

Reviewer 2 Report

Comments and Suggestions for Authors

The paper is interesting however some points could be discussed. The gleason score is one of the best predictors of prostate cancer (PCa) aggressivenes and the molecular test could be a good option as authors mention. The bioinformatic work is well conducted, but the authors don´t  show evidence or validate in other cohort. The heterogeneity is one of the main problems and the paper shows evidence at molecular level in immune cells. The association of GS levels with the activity levels of immunity-related gene sets in PCa could be interesting if is reproducible in other samples or in other cohorts. May be the authors can include in the algorithm PSA levels correlation and age, also diagnosis time. The gene activity needs to be measuring in a different cohort to validate the results. In my opinion the paper needs to be improved, before to publish. 

Author Response

Reviewer 2

The paper is interesting however some points could be discussed. The gleason score is one of the best predictors of prostate cancer (PCa) aggressiveness and the molecular test could be a good option as authors mention. The bioinformatic work is well conducted, but the authors don´t show evidence or validate in other cohort.

The heterogeneity is one of the main problems and the paper shows evidence at molecular level in immune cells. The association of GS levels with the activity levels of immunity-related gene sets in PCa could be interesting if is reproducible in other samples or in other cohorts. May be the authors can include in the algorithm PSA levels correlation and age, also diagnosis time. The gene activity needs to be measuring in a different cohort to validate the results. In my opinion the paper needs to be improved, before to publish.

We thank the reviewer for this thoughtful and constructive feedback. We fully agree that validation in an independent cohort would further strengthen the findings. In accordance with the reviewer's suggestion, we explored additional publicly available datasets and identified the International Cancer Genome Consortium (ICGC) (http://www.icgc.org) prostate cancer cohort, which includes RNA-seq data from 94 patients. Although the sample size and clinical annotation in this cohort were limited, we conducted a similar anaylssis and observed a trend toward a positive association between GS and expression of cell proliferation-related gene sets. While the results did not reach statistically significant due to limited power, the directionality was consistent with our findings in the TCGA and GSE116918 cohorts. This suggests that the transcriptomic signature associated with GS may be reproducible across datasets, pending further validation in larger cohorts.

As our primary aim was to investigate the molecular landscape of GS using RNA-seq data, we did not include clinical parameters such as PSA level, age, or time of diagnosis in the current analysis. This was in part due to the limited availability of public RNA-seq datasets that are sufficiently annotated with these clinical variables. We fully agree, however, that integrating such parameters would improve the clinical applicability of our findings, and we have noted this as a potential direction for future studies in the discussion (Discussion section of page 12, lines 387-389).

Reviewer 3 Report

Comments and Suggestions for Authors

 Title and Abstract

Evaluation:
Clear, appropriate, and aligned with the content of the manuscript.

Comments:

  • The title accurately reflects the scope of the study, highlighting its focus on biological features of prostate cancer with high Gleason scores.

  • The abstract effectively summarizes the background, methodology, key findings, and implications. However, it could benefit from more clarity regarding the specific results of the immune and stromal cell analyses and a mention of the study’s limitations.

 Scientific Rationale and Novelty

Evaluation:
The study addresses a relevant and underexplored topic with clinical implications.

Comments:

  • The paper presents a novel insight by correlating Gleason scores with transcriptomic features, mutation burden, and tumor microenvironment components using in silico analysis.

  • This comprehensive multi-omics integration enhances our understanding of the molecular heterogeneity underlying Gleason grading.

  • It is among the first to link high GS with homologous recombination deficiency (HRD) scores and neoantigen burden, which could inform treatment strategies.

Methodology

Evaluation: Robust, transparent, and well-described.

Comments:

  • The use of GSVA and xCell on two independent large-scale datasets (TCGA and GSE116918) is a strength.

  • The pipeline for assessing immune infiltration, HRD, and mutation load is sound and appropriately referenced (e.g., Thorsson et al.).

  • Minor issues:

    • The term “GS expression” is used inconsistently—it should be clarified that GS is a pathological score, not a gene.

    • Typos such as “Estimaation” and grammatical inconsistencies should be corrected.

 Results

Evaluation:
Findings are clearly presented and statistically supported.

Comments:

  • The results are structured logically: prognosis → proliferation → immune cells → mutation load.

  • The figures are appropriate and enhance data interpretation.

  • The survival analysis is informative; however, results in the RT cohort (GSE116918) are not significant—this should be discussed more explicitly.

Discussion and Interpretation

Evaluation:
Thoughtful and comprehensive.

Comments:

  • The discussion appropriately contextualizes findings with current literature.

  • The authors acknowledge limitations such as retrospective design, single time-point data, and lack of functional validation.

  • The suggestion that high GS tumors may be candidates for PARP inhibitors and immune checkpoint blockade is logical but speculative; more caution is warranted.

 Conclusion

Evaluation:
✔ Concise and well-aligned with the findings.

Comments:

  • The conclusion reiterates the study’s novelty and emphasizes its potential translational impact.

  • The therapeutic implications are interesting but should be framed as hypotheses for future research rather than clinical recommendations.

 References

Evaluation:
Comprehensive and up to date.

Comments:

  • The citation list is well-curated and includes relevant foundational and recent studies.

  • Reference formatting appears consistent with journal requirements.

 Overall Structure, Language, and Clarity

Evaluation:
 Mostly clear, but revision is needed.

Comments:

  • The manuscript would benefit from professional English editing to correct minor grammatical issues, awkward phrasings, and inconsistent terminology (e.g., “GS expression” vs. “GS level”).

  • Flow and readability are generally good, especially in the Methods and Results sections.

  • References: consider in your article this reference:doi: 10.3390/cancers16223766.

Author Response

Reviewer 3

Title and Abstract

Evaluation:

Clear, appropriate, and aligned with the content of the manuscript.

Comments:

  • The title accurately reflects the scope of the study, highlighting its focus on biological features of prostate cancer with high Gleason scores.
  • The abstract effectively summarizes the background, methodology, key findings, and implications. However, it could benefit from more clarity regarding the specific results of the immune and stromal cell analyses and a mention of the study’s limitations.

We thank the reviewer’s comment. Following the word limit of the abstract, the results of the analysis of immune cells and stromal cells with no significant findings have been omitted and we had added new sentences in the manuscript (Discussion section of page 10, lines 333-339).

 Scientific Rationale and Novelty

Evaluation:

The study addresses a relevant and underexplored topic with clinical implications.

Comments:

  • The paper presents a novel insight by correlating Gleason scores with transcriptomic features, mutation burden, and tumor microenvironment components using in silico analysis.
  • This comprehensive multi-omics integration enhances our understanding of the molecular heterogeneity underlying Gleason grading.
  • It is among the first to link high GS with homologous recombination deficiency (HRD) scores and neoantigen burden, which could inform treatment strategies.

We thank the reviewer’s comment. We appreciate your agreement on the novelty of this article.

Methodology

Evaluation: Robust, transparent, and well-described.

Comments:

  • The use of GSVA and xCell on two independent large-scale datasets (TCGA and GSE116918) is a strength.
  • The pipeline for assessing immune infiltration, HRD, and mutation load is sound and appropriately referenced (e.g., Thorsson et al.).
  • Minor issues:
  • The term “GS expression” is used inconsistently—it should be clarified that GS is a pathological score, not a gene.
  • Typos such as “Estimaation” and grammatical inconsistencies should be corrected.

We thank the reviewer’s comment. I have corrected “GS expression” to “GS” and corrected the typo in “Estimaation” to “Estimation”.

 Results

Evaluation:

Findings are clearly presented and statistically supported.

Comments:

  • The results are structured logically: prognosis → proliferation → immune cells → mutation load.
  • The figures are appropriate and enhance data interpretation.
  • The survival analysis is informative; however, results in the RT cohort (GSE116918) are not significant—this should be discussed more explicitly.

We thank the reviewer’s comment. As the reviewer pointed out, the reason why there was no significant difference between BCR-FS and MFS in the radiation therapy cohort could be due to insufficient follow-up period or co-administration of androgen deprivation therapy in high-GS patients treated with radiation therapy. We had added a sentence in the Results section of page 5, lines 194-197.

Discussion and Interpretation

Evaluation:

Thoughtful and comprehensive.

Comments:

  • The discussion appropriately contextualizes findings with current literature.
  • The authors acknowledge limitations such as retrospective design, single time-point data, and lack of functional validation.
  • The suggestion that high GS tumors may be candidates for PARP inhibitors and immune checkpoint blockade is logical but speculative; more caution is warranted.

We thank the reviewer’s comment. As reviewer pointed out, the suggestion that tumors with high GS are candidates for PARP inhibitors or immune checkpoint inhibitors is a speculative comment. We had added a sentence: “However, there is currently no evidence that PARP inhibitors and immune checkpoint inhibitors are effective in High GS levels.” in the Discussion section of page 10, lines 324-325.

 Conclusion

Evaluation:

✔ Concise and well-aligned with the findings.

Comments:

  • The conclusion reiterates the study’s novelty and emphasizes its potential translational impact.
  • The therapeutic implications are interesting but should be framed as hypotheses for future research rather than clinical recommendations.

We thank the reviewer’s comment. In accordance with the reviewer's suggestion, we had added a sentence: “The efficacy of PARP inhibitors and immune checkpoint inhibitors in patients with high GS levels needs to be validated by future clinical studies.” in the Conclusions section of page 12, lines 400-402.

 References

Evaluation:

Comprehensive and up to date.

Comments:

  • The citation list is well-curated and includes relevant foundational and recent studies.
  • Reference formatting appears consistent with journal requirements.

We thank the reviewer’s comment. We appreciate your checking the references for this article.

 Overall Structure, Language, and Clarity

Evaluation:

 Mostly clear, but revision is needed.

Comments:

  • The manuscript would benefit from professional English editing to correct minor grammatical issues, awkward phrasings, and inconsistent terminology (e.g., “GS expression” vs. “GS level”).
  • Flow and readability are generally good, especially in the Methods and Results sections.
  • References: consider in your article this reference:doi: 10.3390/cancers16223766.

We thank the reviewer’s comment. We have checked the article you mentioned (doi: 10.3390/cancers16223766) and cite it in the manuscript (Introduction section of page 3, lines 99-101).

Round 2

Reviewer 1 Report

Comments and Suggestions for Authors

The authors modified the manuscript and responded appropriately point by point to each reviewer's request.

Reviewer 2 Report

Comments and Suggestions for Authors

The authors  has made all changes and  the paper was improved, the paper could be published.